# Conditional Unigram Tokenization with Parallel Data

**Gianluca Vico** [1]  **Jindřich Libovický** [1]

## Abstract

We introduce conditional unigram tokenization, a novel approach that extends unigram tokenization by conditioning target token probabilities on source-language tokens from parallel data. Given a fixed source tokenizer, our method learns a target tokenizer that maximizes cross-lingual semantic alignment. We evaluate our tokenizer on four language pairs across different families and resource levels, examining intrinsic properties and downstream performance on machine translation and language modeling. While our conditional tokenizer maintains comparable statistical properties to standard unigram tokenizers, results are mixed: we observe no improvements in machine translation quality, but find consistent perplexity reductions in language modeling. We hypothesize that quadratic scaling of conditional probability estimation with respect to the vocabulary size creates a data efficiency bottleneck. Our findings suggest that alternative parameterizations may be necessary for practical cross-lingual tokenization.

## 1. Introduction

Tokenization serves as the foundation of most natural language processing pipelines, directly influencing model performance across tasks. While traditional tokenization approaches (Sennrich et al., 2016; Kudo, 2018) focus primarily on token frequency in monolingual contexts, their effectiveness in multilingual scenarios depends critically on achieving both literal (Pires et al., 2019; Limisiewicz et al., 2023) and semantic (Hämmerl et al., 2025) overlap between languages. Improving the semantic overlap of tokenizers in different languages might be beneficial, particularly for low-resource languages that suffer from low performance caused, among others, by overtokenization

[1]Institute of Formal and Applied Linguistics, Faculty of Mathematics and Physics, Charles University, Prague, Czech Republic. Correspondence to: Gianluca Vico <vico [at] ufal.muff.cuni.cz>, Jindřich Libovický <libovicky [at] ufal.muff.cuni.cz>.

*Proceedings of the ICML 2025 Tokenization Workshop (TokShop)*, Vancouver, Canada. PMLR 267, 2025. Copyright 2025 by the author(s).

(Ahia et al., 2023). Therefore, these languages might benefit from cross-lingual alignability.

In this paper, we introduce a novel approach to cross-lingual tokenization that attempts to directly address this challenge in a probabilistic model. Given an existing tokenizer in a source language, we develop a target language tokenizer that maximizes semantic alignment between the two languages. Our approach extends the unigram tokenization framework (Kudo, 2018) by replacing unconditional unigram probabilities with conditional probabilities based on source-language tokens.

Specifically, we formulate tokenization as maximizing the unigram probability of target tokens conditioned on aligned source tokens from parallel data. It is a straightforward generalization of standard unigram tokenization, with the key difference that it explicitly models cross-lingual token alignability during the tokenizer training process. Similarly to the unigram model, this is also used for vocabulary learning.

We evaluate our approach on four language pairs across eight translation directions, analyzing both intrinsic tokenization properties and downstream task performance. Our results present a mixed picture: while the intrinsic evaluation shows that our conditional tokenizer maintains statistical properties comparable to standard unigram tokenizers, we do not observe consistent improvements in machine translation quality. However, we do find notable perplexity reductions in language modeling tasks, suggesting potential benefits for specific applications.

The remainder of this paper is organized as follows: Section 3 details our conditional unigram tokenization approach. Section 4 and 5 present experimental results across multiple language pairs and tasks. Finally, Section 6 discusses implications and directions for future research. The source code for replicating our experiments is openly available on GitHub (https://github.com/GianlucaVico/Conditional-Unigram-Tokenization).

## 2. Related Work

**Subword Tokenization.** The most frequently used subword tokenizers in NLP are BPE (Sennrich et al., 2016) and Unigram (Kudo, 2018). These approaches address out-of-

vocabulary (OOV) words while maintaining a fixed vocabulary size and ensuring tokens have comparable frequencies for proper embedding training. These methods typically represent common words as single tokens, while rare words (including words from low-resource languages, or those in non-Latin scripts) get fragmented into multiple tokens or individual bytes (Petrov et al., 2023; Ahia et al., 2023). Notable alternative approaches include VOLT (Xu et al., 2021), which employs optimal transport for vocabulary construction, or tokenization inference methods, such as PathPiece (Schmidt et al., 2024), which generates the shortest possible token sequence for a given vocabulary, or Legros (Libovický & Helcl, 2024) that finds the most semantically plausible tokenization for a given vocabulary.

**Cross-lingual Token Alignment.** Previous studies (Minixhofer et al., 2022; Remy et al., 2023; 2024) showed that token semantic similarity across languages is important for effective cross-lingual transfer. This similarity can be derived from bilingual dictionaries (Minixhofer et al., 2022) or through automated techniques (Remy et al., 2024), such as Fast Align (Dyer et al., 2013). Hämmerl et al. (2025) establish that token alignment between parallel sentences correlates with performance on multiple downstream tasks and introduces metrics for measuring such alignment across different tokenizers using a statistical model for word alignment.

**Joint Tokenization and Alignment.** Several approaches integrate alignment considerations into tokenization. Chung & Gildea (2009) propose using word alignment between parallel sentences for Chinese word segmentation. While their approach shares similarities with our work through its foundation in word alignment, key differences exist: (1) they derive tokenization from alignment, whereas we compute tokenization directly with alignment as a by-product, and (2) they use an explicit hyperparameter to control tokenized sequence compression, while in our method, compression emerges naturally from the algorithm.

Deguchi et al. (2020) developed a machine translation-specific tokenization method that selects subword segmentations of parallel sentences to maximize unigram language model probability while maintaining similar length. This approach aims at better efficiency and reaches better text compression without sacrificing tokenization quality, but does not optimize for semantic overlap.

**Word Alignment Methods.** The word alignment field includes statistical approaches such as the IBM models (Brown et al., 1993) and Eflomal (Östling & Tiedemann, 2016), as well as neural network-based methods like Awesome Align (Dou & Neubig, 2021). These tools focus on the alignment task rather than integrating it with tokenization.

## 3. Alignable Tokenization

For cross-lingually alignable tokenization, we assume a fixed tokenizer for the source language and access to parallel data between the source and target languages. The goal is to derive a target-language tokenization such that subwords in both languages are semantically aligned. Moreover, we require that it is possible to reconstruct the original text by simply concatenating the tokens (and removing some special characters). We adopt a probabilistic formulation similar to the Unigram tokenizer, but condition token probabilities on the fixed source language tokenization:

$$\mathcal{L}(T, S) = \underset{\text{Tok}}{\text{argmax}} \sum_{t \in \text{Tok}(T)} - \log p(t \mid S) \qquad (1)$$

where $\text{Tok}$ is a function that splits the target-language sequence $T$ into tokens, and $S$ is a source-language sequence encoded as tokens. $T$ is the translation of $S$. The objective is to find target-language character spans that align with source-language tokens.

Estimating $p(t \mid S)$ directly is intractable. We simplify it by treating the source sentence as a bag of tokens and computing the probability as:

$$p(t \mid S) = \frac{p(t, S)}{p(S)} \approx \frac{\sum_{s_i \in S} c(t, s_i)}{\sum_{t_j \in V_{\text{tgt}}} \sum_{s_k \in S} c(t_j, s_k)} \qquad (2)$$

where $c(t, s)$ counts the co-occurrences of tokens $t$ and $s$ in sequence pairs in a corpus containing parallel sentences, and $V_{\text{tgt}}$ represents the target vocabulary.

Given $p(t \mid S)$, we find a segmentation that maximizes the overall probability using the Unigram model's dynamic programming algorithm. Initially, the vocabulary $V_{\text{tgt}}$ contains all character spans from the training data (up to a fixed length), and $c(t, s)$ is estimated based on all the possible tokens in the target language and tokens in the source language. At every iteration, we update it by computing the expected number of co-occurrences and using only the target tokens currently in the vocabulary $V_{\text{tgt}}$. For a particular training example $T$, this is proportional to the probability of observing the prefix ($T_{:i}$), the token itself ($T_{i:j}$), and the suffix ($T_{j:}$). Then, the amount is distributed across the source tokens, so that the contribution of a pair of tokens ($T_{i:j}, s$) from the training example ($T, S$) is the following:

$$c_{\text{sample}}(T_{i:j}, s) = \frac{p(T_{i:j} \mid S) \; p(T_{:i} \mid S) \; p(T_{j:} \mid S)}{\text{length}(S)} \qquad (3)$$

Then, these quantities are accumulated to obtain the updated count table.

We experiment also with an alternative training method similar to expectation maximization, where we iterate the

following two steps: First, after initializing the table $c(t, s)$, we use it to tokenize the text; Second, we use the tokenized text to update the table by increasing the count of the tokens that appear in it. However, with this method, tokens that do not appear during the first iteration are never counted and so they are immediately removed from the vocabulary. For this reason, in our experiments, we will compare both methods, but focus mostly on the former one.

With either training method, we initialize the target vocabulary with all character spans up to a fixed length. Similarly to the unigram model, we reduce the vocabulary iteratively, always after adjusting the unigram probabilities. We keep the subwords with the highest mutual information with the source tokens until the desired vocabulary size is reached. In this way, we can penalize pairs of tokens where one of them is rare while the other is frequent and that appear together by chance. Single characters are always kept in the vocabulary.

$$I(t, V_{\text{src}}) = \sum_{s \in V_{\text{src}}} p(t, s) \log \frac{p(t, s)}{p(t)p(s)} \tag{4}$$

To reduce the memory requirements and speed up the training, we pretokenize the input sentences and use Eflomal (Östling & Tiedemann, 2016) to align the words. Then, each pair of aligned words is used as a training example instead of the full sentences. Although we do not experiment with languages without white spaces, this step can be skipped entirely or adapted to such languages by using a different pre-tokenization method.

Token alignment probabilities between two tokens $t$ and $s$ can be computed as:

$$p(t \mid s) = \frac{c(t, s)}{\sum\limits_{t_i \in V_{\text{tgt}}} c(t_i, s)} \tag{5}$$

For a given target sequence, we consider only tokens that are substrings of the target sequence in the denominator instead of the entire vocabulary $V_{\text{tgt}}$.

This formulation requires both source and target sequences for target tokenization. Alternatively, only the target sequence can be used by estimating $p(t)$ via marginalization:

$$p(t) = \sum_{s \in V_{\text{src}}} p(t, s) = \frac{\sum\limits_{s_i \in V_{\text{src}}} c(t, s_i)}{\sum\limits_{t_j \in V_{\text{tgt}}} \sum\limits_{s_k \in V_{\text{src}}} c(t_j, s_k)} \tag{6}$$

where $V_{\text{src}}$ is the vocabulary of the fixed source tokenizer. This resembles an unconditional Unigram tokenizer but with tokens counted differently. Alternatively, following Libovický & Helcl (2024), we can use the tokenized text to distill a bigram model.

The simplified pseudo-code for training our tokenizer is shown in Appendix E.

## 4. Experiments

First, we evaluate our model intrinsically and then on two tasks: machine translation, since it requires parallel data, and language modeling to investigate its performance without parallel data.

We focus on the following language pairs:

**French (fra) & Italian (ita).** Both languages are high resources (Tier 5 and 4 according to Joshi et al., 2020) from the same family and use the same alphabet.

**Czech (ces) & Ukrainian (ukr).** Compared to the previous pair, this is a less-resourced language pair (Tier 4 and 3). The languages are from the same family but use different scripts.

**Italian (ita) & Maltese (mlt).** They differ in families but share the same script. Maltese, a low-resource Semitic language (Tier 2), has complex morphology with infixes but shows Italian influence due to geographical proximity.

**German (deu) & Upper Sorbian (hsb).** Both languages are spoken in Germany, but they come from different families. German is a high-resource language (Tier 5), while Upper Sorbian is a low-resource Slavic language (Tier 1).

For French-Italian and Czech-Ukrainian, we train the tokenizers with 100k, 500k, and 1M examples. The data is from NLLB (NLLB Team et al., 2022), which contains 47M examples for French-Italian and 4M for Czech-Ukrainian. For Italian-Maltese, we use 100k examples from Multi-ParaCrawl (Bañón et al., 2020), which totals 483k examples. For German-Upper Sorbian, we use 60k examples from WMT2020 (Libovický & Fraser, 2021).

We use Flores (NLLB Team et al., 2022) for evaluating the tokenizers, with the exception of German-Upper Sorbian, which is evaluated on the WMT2020 test set.

### 4.1. Intrinsic Evaluation

We compare our tokenizers against Unigram models from SentencePiece trained on identical data with matching vocabulary sizes (8k, 16k, 32k). These baseline models also serve as the source tokenizers for training our conditional tokenizers. We use the following notation: $\text{SP}_{\text{src}}$ and $\text{SP}_{\text{tgt}}$ refer to SentencePiece tokenizers for source and target languages, respectively (e.g., for the pair Czech $\rightarrow$ Ukrainian, $\text{SP}_{\text{src}}$ is trained on Czech, $\text{SP}_{\text{tgt}}$ on Ukrainian), while PairedSP refers to *Paired SentencePiece*. We also evaluate two variants: PairedSP trained with Expectation

Table 1: Parity scores of the different tokenizers when trained with the largest training set available for the language pairs.

| | | Parity ($\downarrow$) | | | | | |
|---|---|---|---|---|---|---|---|
| Size | Model | 8k | 16k | 32k | 8k | 16k | 32k |
| | | fra $\to$ ita | | | ita $\to$ fra | | |
| 1m | PairedSP | 1.24 | 1.11 | 1.04 | 1.22 | 1.16 | 1.13 |
| | PairedSP$_M$ | 3.95 | 1.07 | 0.99 | 1.19 | 1.11 | 1.08 |
| | PairedSP$_{EM}$ | 1.06 | 1.07 | 1.04 | 1.16 | 1.16 | 1.16 |
| | SP$_{tgt}$ | **0.96** | **0.95** | **0.95** | **1.04** | **1.05** | **1.05** |
| | | ces $\to$ ukr | | | ukr $\to$ ces | | |
| 1m | PairedSP | 1.59 | 1.51 | 1.39 | 1.41 | 1.30 | 1.18 |
| | PairedSP$_M$ | 1.58 | 1.49 | 1.36 | 1.39 | 1.26 | 1.13 |
| | PairedSP$_{EM}$ | 1.11 | 1.15 | 1.16 | 1.05 | 1.07 | 1.05 |
| | SP$_{tgt}$ | **1.02** | **1.03** | **1.05** | **0.98** | **0.97** | **0.95** |
| | | ita $\to$ mlt | | | mlt $\to$ ita | | |
| 100k | PairedSP | 1.43 | 1.28 | 1.20 | 1.21 | 1.09 | 1.00 |
| | PairedSP$_M$ | 1.41 | 1.24 | 1.15 | 1.19 | 1.04 | 0.95 |
| | PairedSP$_{EM}$ | 1.16 | 1.17 | 1.13 | 0.99 | 0.99 | 0.95 |
| | SP$_{tgt}$ | **1.08** | **1.08** | **1.09** | **0.93** | **0.92** | **0.92** |
| | | deu $\to$ hsb | | | hsb $\to$ deu | | |
| 60k | PairedSP | 1.37 | 1.20 | 1.07 | 1.32 | 1.20 | 1.04 |
| | PairedSP$_M$ | 1.35 | 1.18 | 1.05 | 1.32 | 1.18 | 1.08 |
| | PairedSP$_{EM}$ | 1.03 | 1.01 | **0.95** | 1.05 | 1.05 | **1.01** |
| | SP$_{tgt}$ | **1.00** | **0.99** | 0.98 | **1.00** | **1.01** | 1.02 |

Table 2: Fertility scores of the different tokenisers when trained with the largest training set available for the language pairs.

| | | Fertility ($\downarrow$) | | | | | |
|---|---|---|---|---|---|---|---|
| Size | Model | 8k | 16k | 32k | 8k | 16k | 32k |
| | | fra $\to$ ita | | | ita $\to$ fra | | |
| 1m | PairedSP | 1.76 | 1.43 | 1.26 | 1.52 | 1.30 | 1.19 |
| | PairedSP$_M$ | 5.61 | 1.37 | 1.20 | 1.48 | 1.25 | 1.14 |
| | PairedSP$_{EM}$ | 1.51 | 1.38 | 1.27 | 1.45 | 1.31 | 1.22 |
| | SP$_{tgt}$ | **1.37** | **1.23** | **1.15** | **1.30** | **1.18** | **1.11** |
| | | ces $\to$ ukr | | | ukr $\to$ ces | | |
| 1m | PairedSP | 2.61 | 2.14 | 1.76 | 2.46 | 1.99 | 1.64 |
| | PairedSP$_M$ | 2.59 | 2.12 | 1.72 | 2.42 | 1.93 | 1.56 |
| | PairedSP$_{EM}$ | 1.82 | 1.63 | 1.46 | 1.83 | 1.64 | 1.45 |
| | SP$_{tgt}$ | **1.67** | **1.47** | **1.33** | **1.71** | **1.48** | **1.32** |
| | | ita $\to$ mlt | | | mlt $\to$ ita | | |
| 100k | PairedSP | 1.82 | 1.48 | 1.32 | 1.87 | 1.55 | 1.36 |
| | PairedSP$_M$ | 1.79 | 1.44 | 1.27 | 1.83 | 1.48 | 1.29 |
| | PairedSP$_{EM}$ | 1.47 | 1.36 | 1.25 | 1.53 | 1.41 | 1.28 |
| | SP$_{tgt}$ | **1.37** | **1.26** | **1.20** | **1.43** | **1.31** | **1.25** |
| | | deu $\to$ hsb | | | hsb $\to$ deu | | |
| 60k | PairedSP | 2.22 | 1.79 | 1.53 | 1.96 | 1.62 | 1.33 |
| | PairedSP$_M$ | 2.20 | 1.76 | 1.50 | 1.95 | 1.60 | 1.38 |
| | PairedSP$_{EM}$ | 1.67 | 1.51 | **1.35** | 1.55 | 1.41 | **1.28** |
| | SP$_{tgt}$ | **1.63** | **1.48** | 1.40 | **1.48** | **1.36** | 1.30 |

Maximization (PairedSP$_{EM}$), and a version that tokenizes only target sequences without source context (PairedSP$_M$). Note that PairedSP and PairedSP$_M$ share identical parameters (the co-occurrence table $c(t, s)$) but differ in their tokenization procedures.

We assess tokenization quality using the following metrics:

**Parity ($\downarrow$).** This measures the ratio of tokens produced by our tokenizer in the target language to those produced by the reference tokenizer in the source language (Petrov et al., 2023). Optimal tokenization should yield similar sequence lengths across languages.

**Fertility ($\downarrow$).** This measures the average number of tokens per word (Rust et al., 2021). Lower fertility (minimum 1.0) indicates that words remain coherent semantic units.

For alignment quality assessment, we first get the token alignment on the test data using Eflomal and we compare PairedSP and SP$_{tgt}$ using:

**One-to-one ($\uparrow$).** Following Hämmerl et al. (2025), this measures the proportion of source tokens that have exactly one aligned target token which is also aligned to exactly one token. We measure this on the source side due to its fixed tokenization.

**Unaligned ($\downarrow$).** It is the portion of source tokens that are not aligned to any target tokens. As for *One-to-one*, we measure it on the source sequence.

We tokenize the dev tests with both SP$_{tgt}$ and PairedSP, then mark the tokens to recognize which tokenizer produced them. After joining the two sets, we train Eflomal to align this set in the target language to the one in the source language, tokenized by SP$_{src}$. We prepare the test sets in the same way, and we use them to compute the alignment metrics with the Eflomal priors computed on the dev tests. In this way, Eflomal can align sentences tokenized by either model, and we can compare the metrics computed for both tokenizers.

### 4.2. Machine Translation

We evaluate our tokenizer on machine translation, hypothesizing that improved token correspondence between languages should simplify MT model training by making the task more similar to token-level translation rather than complex sequence-to-sequence mapping.

We use the same language pairs and tokenizers as in intrinsic evaluation, testing three vocabulary sizes with the largest available training set for each language pair. Our experimental setup uses SP$_{src}$ for input tokenization and PairedSP

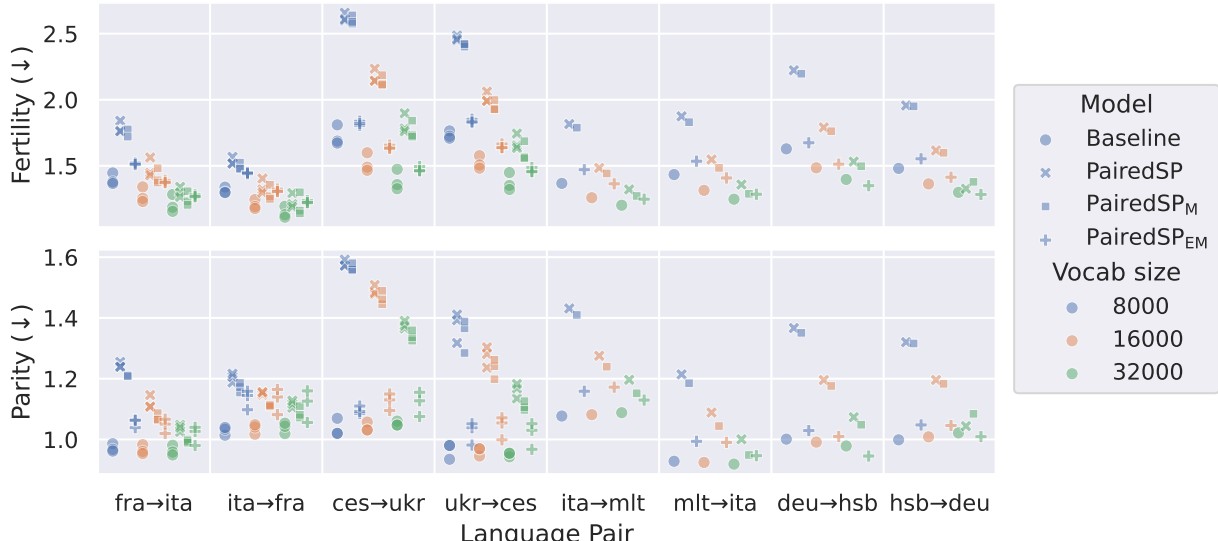

Figure 1: Fertility and parity scores of the tokenizers on the different language pairs, subdivided by vocabulary size (color) and model (shape). There is an outlier (PairedSP$_M$ fra $\rightarrow$ ita 8k vocabulary size) that is not shown for clarity: it has 5.61 fertility and 3.95 parity.

Table 3: One-to-one scores of the tokenizers trained on the largest training set available for each language.

| | | One-to-one ($\uparrow$) | | | | | |
|---|---|---|---|---|---|---|---|
| Size | Model | 8k | 16k | 32k | 8k | 16k | 32k |
| | | fra $\rightarrow$ ita | | | ita $\rightarrow$ fra | | |
| 1m | PairedSP | 0.55 | 0.59 | 0.60 | 0.59 | **0.62** | 0.62 |
| | SP$_{tgt}$ | **0.58** | **0.60** | **0.61** | **0.60** | **0.62** | **0.63** |
| | | ces $\rightarrow$ ukr | | | ukr $\rightarrow$ ces | | |
| 1m | PairedSP | 0.49 | 0.58 | 0.60 | 0.52 | 0.58 | 0.61 |
| | SP$_{tgt}$ | **0.59** | **0.62** | **0.63** | **0.58** | **0.60** | **0.61** |
| | | ita $\rightarrow$ mlt | | | mlt $\rightarrow$ ita | | |
| 100k | PairedSP | 0.45 | 0.52 | 0.53 | 0.47 | 0.50 | 0.50 |
| | SP$_{tgt}$ | **0.53** | **0.53** | **0.54** | **0.51** | **0.51** | **0.51** |
| | | deu $\rightarrow$ hsb | | | hsb $\rightarrow$ deu | | |
| 60k | PairedSP | 0.64 | 0.66 | 0.67 | 0.66 | 0.68 | 0.69 |
| | SP$_{tgt}$ | **0.67** | **0.68** | **0.70** | **0.67** | **0.70** | **0.72** |

Table 4: Unaligned scores of the tokenizers trained on the largest training set available for each language.

| | | Unaligned ($\downarrow$) | | | | | |
|---|---|---|---|---|---|---|---|
| Size | Model | 8k | 16k | 32k | 8k | 16k | 32k |
| | | fra $\rightarrow$ ita | | | ita $\rightarrow$ fra | | |
| 1m | PairedSP | **0.18** | **0.21** | **0.21** | **0.17** | **0.17** | **0.16** |
| | SP$_{tgt}$ | 0.23 | 0.22 | 0.22 | 0.20 | 0.18 | 0.18 |
| | | ces $\rightarrow$ ukr | | | ukr $\rightarrow$ ces | | |
| 1m | PairedSP | **0.13** | **0.17** | **0.19** | **0.17** | **0.21** | **0.22** |
| | SP$_{tgt}$ | 0.23 | 0.22 | 0.20 | 0.25 | 0.24 | 0.24 |
| | | ita $\rightarrow$ mlt | | | mlt $\rightarrow$ ita | | |
| 100k | PairedSP | **0.16** | **0.18** | **0.20** | **0.21** | **0.26** | 0.28 |
| | SP$_{tgt}$ | 0.22 | 0.20 | 0.20 | 0.27 | 0.27 | **0.27** |
| | | deu $\rightarrow$ hsb | | | hsb $\rightarrow$ deu | | |
| 60k | PairedSP | **0.21** | 0.24 | 0.26 | **0.20** | 0.22 | 0.23 |
| | SP$_{tgt}$ | 0.23 | **0.22** | **0.22** | 0.23 | **0.21** | **0.20** |

for output tokenization, with SP$_{tgt}$ replacing PairedSP as the baseline.

We train the models using Marian (Junczys-Dowmunt et al., 2018) with the Transformer-base architecture (Vaswani et al., 2017) (hyperparameter details in Appendix F). Each model undergoes 1M training updates using data from NLLB, MultiParaCrawl, or WMT2020, depending on the language pair.

We evaluate models using chrF++ on Flores test sets (and

WMT2020 test set for German-Upper Sorbian), and additionally report BLEU, TER from SacreBLEU (Post, 2018), and COMET scores (Rei et al., 2020). Complete details are provided in Appendices D and I.

### 4.3. Language Modeling

Finally, we evaluate our tokenizer in a setting without access to parallel data during inference. We train small GPT-2-like models (Radford et al., 2019) (91M to 110M parameters

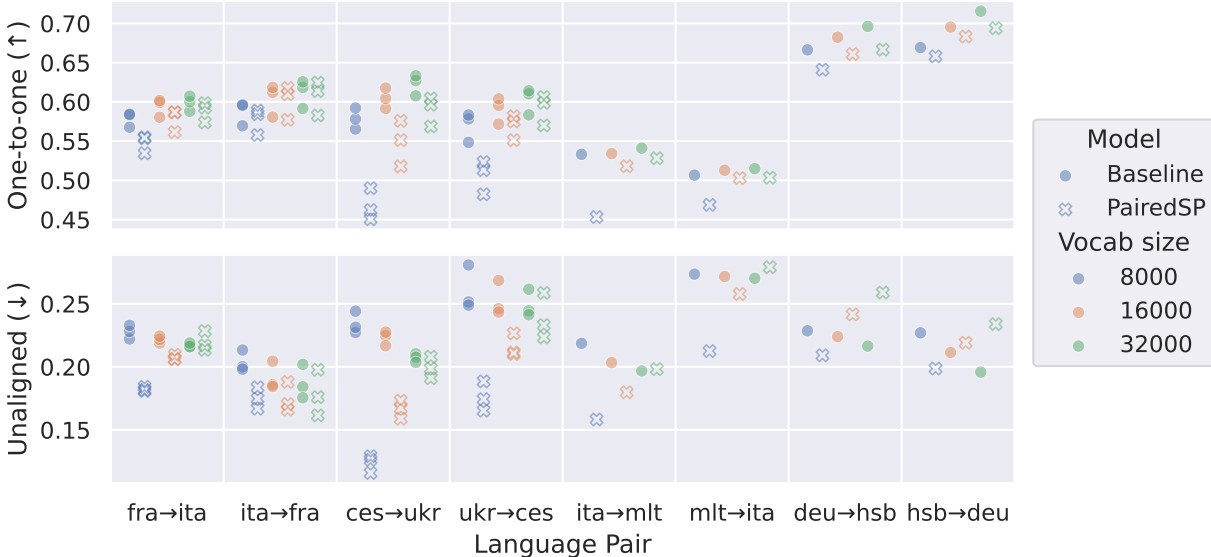

Figure 2: Alignment scores of the tokenizers on the different language pairs subdivided by vocabulary size.

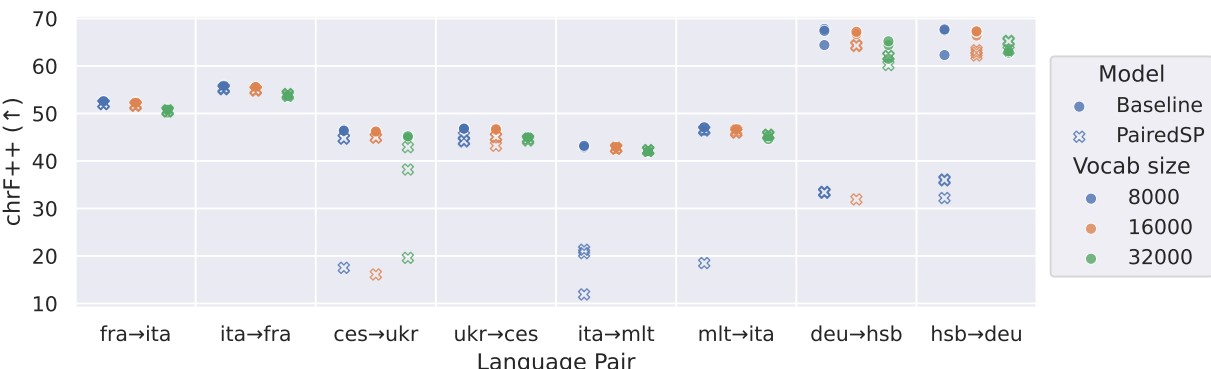

Figure 3: chrF++ (↑) scores on the different language pairs and vocabulary sizes. For most pairs, the baseline has higher scores than PairedSP and lower variance.

depending on the vocabulary size) from scratch using the HuggingFace implementation on the target language of each language pair (hyperparameter details in Appendix G).

We compare two settings, monolingual and bilingual, and each model is trained on a fixed number of examples (2M) to ensure fair comparison. We compare PairedSP$_M$ against SP$_{tgt}$ as the baseline. Importantly, while monolingual models observe only monolingual data during training, the PairedSP$_M$ tokenizer was trained with cross-lingual support from SP$_{src}$, allowing us to assess whether cross-lingual tokenizer training benefits monolingual language modeling. Models are tested only on the target language, and we use perplexity per byte to compare models with different vocabularies.

## 5. Results & Discussion

### 5.1. Intrinsic Evaluation

Figure 1 presents the intrinsic tokenization metrics. The baseline consistently outperforms PairedSP and its variants on both parity and fertility metrics, though this difference diminishes with larger vocabulary sizes. PairedSP$_M$ shows comparable performance to PairedSP, indicating that marginalization does not substantially impact performance in most cases. However, there is one notable failure case: with French-Italian using 1M training examples and 8k vocabulary size, PairedSP$_M$ produces only single-character tokens, resulting in fertility and parity scores of 5.61 and 3.95, respectively.

As expected, larger vocabulary sizes generally improve

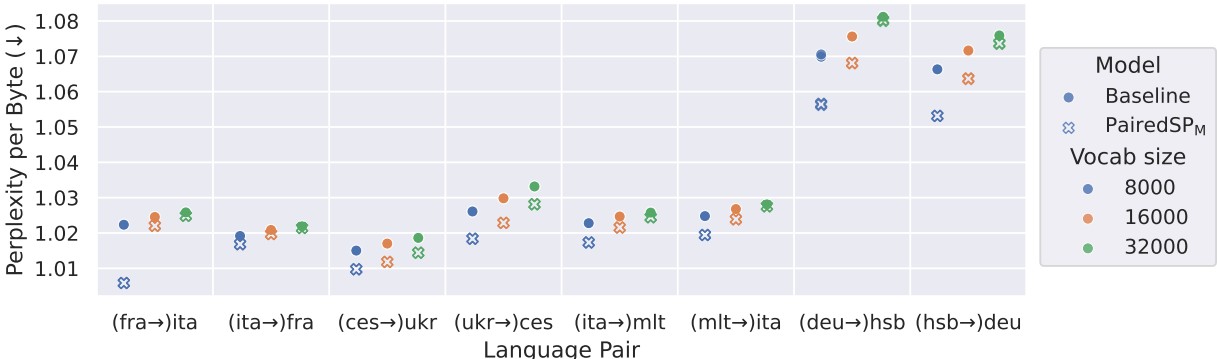

Figure 4: Perplexity per byte of bilingual language models trained on the different languages, subdivided by vocabulary size. PairedSP$_M$ has generally better scores than the baseline, and the models have less variance than in the MT task. In parentheses, there is the source language used to train PairedSP.

Table 5: Average chrF++ scores on the different language pairs and vocabulary sizes.

| | **chrF++ ($\uparrow$)** | | | | | |
|---|---|---|---|---|---|---|
| Model | 8k | 16k | 32k | 8k | 16k | 32k |
| | **fra $\rightarrow$ ita** | | | **ita $\rightarrow$ fra** | | |
| SP$_{src}$ + PairedSP | 52.0 | 51.7 | 50.5 | 55.2 | 54.9 | **53.9** |
| SP$_{src}$ + SP$_{tgt}$ | **52.5** | **52.3** | **50.6** | **55.8** | **55.5** | 53.7 |
| | **ces $\rightarrow$ ukr** | | | **ukr $\rightarrow$ ces** | | |
| SP$_{src}$ + PairedSP | 35.7 | 35.3 | 33.6 | 44.6 | 44.4 | 44.5 |
| SP$_{src}$ + SP$_{tgt}$ | **46.4** | **46.1** | **45.0** | **46.9** | **46.6** | **44.9** |
| | **ita $\rightarrow$ mlt** | | | **mlt $\rightarrow$ ita** | | |
| SP$_{src}$ + PairedSP | 17.9 | 42.7 | **42.2** | 37.2 | 46.1 | **45.4** |
| SP$_{src}$ + SP$_{tgt}$ | **43.0** | **42.8** | 42.0 | **46.8** | **46.5** | 45.1 |
| | **deu $\rightarrow$ hsb** | | | **hsb $\rightarrow$ deu** | | |
| SP$_{src}$ + PairedSP | 33.4 | 53.5 | 61.4 | 34.7 | 62.8 | **64.6** |
| SP$_{src}$ + SP$_{tgt}$ | **66.5** | **67.0** | **63.7** | **65.8** | **67.0** | 62.9 |

Table 6: Average Perplexity per Byte ($\downarrow$) scores on the different language pairs and vocabulary size.

| | | **Perplexity per Byte ($\downarrow$)** | | | | | |
|---|---|---|---|---|---|---|---|
| Model | Setting | 8k | 16k | 32k | 8k | 16k | 32k |
| | | **(fra $\rightarrow$) ita** | | | **(ita $\rightarrow$) fra** | | |
| PairedSP$_M$ | Mono | **1.006** | **1.021** | **1.024** | **1.016** | **1.019** | **1.020** |
| PairedSP$_M$ | Bi | 1.006 | 1.022 | 1.025 | 1.017 | 1.020 | 1.021 |
| SP$_{tgt}$ | Mono | 1.021 | 1.023 | 1.025 | 1.018 | 1.020 | 1.021 |
| SP$_{tgt}$ | Bi | 1.022 | 1.025 | 1.026 | 1.019 | 1.021 | 1.022 |
| | | **(ces $\rightarrow$) ukr** | | | **(ukr $\rightarrow$) ces** | | |
| PairedSP$_M$ | Mono | **1.009** | **1.011** | **1.014** | **1.017** | **1.022** | **1.027** |
| PairedSP$_M$ | Bi | 1.010 | 1.012 | 1.014 | 1.018 | 1.023 | 1.028 |
| SP$_{tgt}$ | Mono | 1.014 | 1.016 | 1.018 | 1.025 | 1.028 | 1.031 |
| SP$_{tgt}$ | Bi | 1.015 | 1.017 | 1.019 | 1.026 | 1.030 | 1.033 |
| | | **(ita $\rightarrow$) mlt** | | | **(mlt $\rightarrow$) ita** | | |
| PairedSP$_M$ | Mono | 1.018 | 1.023 | 1.026 | 1.020 | 1.025 | 1.029 |
| PairedSP$_M$ | Bi | **1.017** | **1.022** | **1.024** | **1.019** | **1.024** | **1.028** |
| SP$_{tgt}$ | Mono | 1.024 | 1.026 | 1.027 | 1.026 | 1.028 | 1.030 |
| SP$_{tgt}$ | Bi | 1.023 | 1.025 | 1.026 | 1.025 | 1.027 | 1.028 |
| | | **(deu $\rightarrow$) hsb** | | | **(hsb $\rightarrow$) deu** | | |
| PairedSP$_M$ | Mono | 1.063 | 1.074 | 1.085 | 1.059 | 1.069 | 1.078 |
| PairedSP$_M$ | Bi | **1.056** | **1.068** | **1.080** | **1.053** | **1.064** | **1.074** |
| SP$_{tgt}$ | Mono | 1.077 | 1.081 | 1.085 | 1.073 | 1.078 | 1.080 |
| SP$_{tgt}$ | Bi | 1.070 | 1.076 | 1.081 | 1.066 | 1.072 | 1.076 |

these metrics. Contrary to our expectations, PairedSP$_{EM}$ performs better than PairedSP despite not updating counts for rare tokens, though it still underperforms the baseline. Additionally, as shown in Tables 1 and 2, PairedSP$_M$ outperforms PairedSP. We hypothesize that this occurs because PairedSP$_M$'s probability estimation more closely resembles that of SP$_{tgt}$.

We observe similar patterns in the one-to-one alignment metric. PairedSP shows improvement over the baseline on the unaligned metric, indicating that it leaves fewer source tokens without target alignments. While larger vocabulary sizes improve the one-to-one metric consistently, they improve the unaligned metric only for the baseline, possibly due to increased difficulty in estimating the $c(t, s)$ table that is quadratically bigger compared to the unconditional

probability $p(t)$ in the Unigram model.

### 5.2. Machine Translation

Figure 3 shows chrF++ scores for machine translation models across language pairs and vocabulary sizes. The baseline consistently outperforms our model. For some language pairs like French-Italian, the difference is minimal (0.33 chrF++ on average), while for others like Czech-Ukrainian, it is more substantial (6.31 chrF++ on average). Vocabulary

size appears to have minimal effect on results, with a slight decrease in scores for larger vocabularies given the same number of training steps.

Table 5 shows that PairedSP outperforms the baseline in only four cases on average. Notably, Czech-Ukrainian shows the largest performance decrease when using PairedSP, though this pattern does not hold in the reverse direction.

Furthermore, scores with our tokenizer exhibit much higher variance than the baseline (except for French-Italian), suggesting that this tokenization approach may be less reliable than standard SentencePiece.

### 5.3. Language Modeling

Figure 4 demonstrates that PairedSP$_M$ achieves improved perplexity per byte across all language pairs and vocabulary sizes compared to the baseline. Interestingly, this improvement does not correlate with tokenization scores: the PairedSP$_M$ model with the worst intrinsic evaluation scores achieves the lowest perplexity in language modeling. Table 6 shows that bilingual training with both the source and target language improves the perplexity per byte on the target language in low-resource languages.

## 6. Conclusions

We presented a novel tokenization method that leverages parallel data to improve cross-lingual token alignment. Our approach extends the unigram tokenization framework by conditioning target token probabilities on source language tokens, with the goal of achieving better semantic alignment between languages.

Our experimental evaluation reveals mixed results across different tasks and metrics. While our method does not consistently improve intrinsic tokenization metrics or machine translation quality compared to standard unigram tokenizers, we observe consistent perplexity reductions in language modeling tasks.

We hypothesize that the performance gap between our approach and standard unigram tokenization stems from the increased memory complexity of the underlying estimation problem: while a table storing $p(t)$ scales linearly with vocabulary size, $p(t \mid S)$ scales quadratically, yet the available training data remains the same. This scaling issue may particularly impact low-resource languages, contrary to our initial motivation.

Based on our observations, we estimate that approximately 28M examples would be required to match unigram fertility and 4M examples for comparable one-to-one alignment performance. These requirements may limit the practical applicability of our approach, especially for the low-resource

scenarios where improved tokenization is most needed.

Future work should explore more data-efficient methods for learning cross-lingually aligned tokenizations. Potential directions include investigating alternative parameterizations that scale more favorably with vocabulary size, exploring pre-training strategies that leverage multilingual representations, or developing hybrid approaches that combine the benefits of both conditional and unconditional tokenization methods.

## Impact Statement

The aim of this paper is to contribute to diminishing the gap between high- and low-resource languages. However, it investigates only a limited number of languages and from a limited geographical area. We do not see any direct ethical risk related to this work.

## Acknowledgements

We thank Martin Popel for comments on the draft of this paper. This research was supported by the Czech Science Foundation project 25-16242S.

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

## A. Examples

The Tables 7 8 and 9 show respectively tokenization, machine translation, and language modelling examples.

Table 7: Tokenization examples from the different models on two specific settings. Tokens are separated by a white space. "_" denotes a white space in the original sentence, which can be reconstructed by concatenating the tokens.

| **Fra → Ita 8k, vocabulary, 1M training set** | |
|---|---|
| $SP_{src}$ (ref.) | _« _Nous _avons _à _présent _des _souri s _de _4 _mois [...] |
| $SP_{tgt}$ | _" _Abbiam o _top i _di _quattro _mesi [...] |
| PairedSP | _" _Abbiamo _to p i _di _quattro _mesi [...] |
| $PairedSP_{EM}$ | _" _Abbiamo _ to pi _di _quattro _mesi [...] |
| $PairedSP_M$ | _ " _ A b b i a m o _ t o p i _ d i _ q u a t t r o _ m e s I [...] |
| **Ces → Ukr, 32k vocabulary, 500k training set** | |
| $SP_{src}$ (ref.) | _" _Зараз _у _нас _є _4 _- _місячні _миші |
| $SP_{tgt}$ | _„ _Nyn í _má me _čtyř měsíční _myši |
| PairedSP | _„ _Nyní _máme _čtyř mě s í č ní _myši |
| $PairedSP_{EM}$ | _„ _Nyní _máme _čtyř měsíční _myš i |
| $PairedSP_M$ | _„ _Nyní _máme _čtyř mě s í č ní _myši |

*"We (now) have four-month-old mice [...]*

Table 8: Machine translation examples from two specific settings. The output of the model is shown tokenized.

| **Czech → Ukrainian, 8k vocabulary** | |
|---|---|
| Source | „Nyní máme čtyřměsíční myši bez cukrovky, které ji dříve měly," dodal. |
| Target | "Зараз у нас є 4-місячні миші, в яких немає діабету і які мали діабет раніше,"— додав він. |
| $SP_{src}$ +$SP_{tgt}$ | _" _Зараз _у _нас _чотири місячн а _ми ша _без _діабет у _, _яка _раніше _була _у _неї _" _, _- _додав _він _. |
| $SP_{src}$ +PairedSP | _" _Зараз _у _нас _є _чотири м і с я ч н і _ми ш і _без _д і а б е т у _, _які _раніше _мали _" _, _- _додав _він _. |

*"We now have 4-month-old mice that are non-diabetic that used to be diabetic," he added.*

| **Upper Sorbian → German, 32k vocabulary** | |
|---|---|
| Source | To njepłaći jenož za naše měšćanske zarjadnišća. |
| Target | Das gilt nicht nur für unsere städtischen Verwaltungen. |
| $SP_{src}$ +$SP_{tgt}$ | _Dies _g ilt _nicht _ nur _für _unsere _unsere _städtische n _Einrichtung en _. |
| $SP_{src}$ +PairedSP | _Das _gilt _nicht _nur _für _unsere _Stadtverwaltung _. |

*This does not only apply to our municipal administrations.*

## B. Tokenizer preprocessing

We use a character coverage of 1.0 and normalize the text with NFKC to reduce differences with the SentencePiece implementation. Moreover, we prepend whitespace to punctuation characters and to the beginning of the sentence. Then, whitespaces are replaced with U+2581.

The relevant SentencePiece settings are the following:

- character coverage: 1.0
- shrinking_factor: 0.75
- num_sub_iterations: 2
- allow_whitespace_only_pieces: true

Table 9: Language modelling examples from two specific settings. The output of the model is shown tokenized. The prediction of model, computed in an autoregressive way, is shown in bold.

| | (Italian → )French, 16k vocabulary |
|---|---|
| SP$_{tgt}$ | _« _Nous _avons _à _présent _des _souris _de _4 _mois **_qui _ont _été _infecté es _par _le _virus _de _la** |
| PairedSP | _« _Nous _avons _à _présent _des _souris _de _4 _mois **_et _des _souris _de _plus _de _6 _mois _, _mais** |

| | (German → )Upper Sorbian, 32k vocabulary |
|---|---|
| SP$_{tgt}$ | _To _njepłaći _je nož _za _naše _měšćanske _zarjadnišća _.  **e _, _ kotrež _ma my _tu _ja ra _dołh** |
| PairedSP | _To _njepłaći _jenož _za _naše _měšćanske _zarjadnišća _.  ** _, _kotrež _wustawki _Domowiny _, _kotryž _je _za ł o** |

- byte_fallback: true

We use equivalent settings for PairedSP.

## C. Metrics for the Intrinsic Evaluation

This is a list of additional metrics we considered for the intrinsic evaluation for the tokenization task:

**Single Character tokens.** We count the proportion of tokens in a sequence that are just a single character.

**Vocabulary usage.** We compute the proportion of tokens in the vocabulary that is actually used on the test set.

**Vocabulary overlap.** This is the portion of vocabulary that overlaps between our tokenizer and the reference one. Both tokenizers are trained on the same languages.

**Length ratio with respect to SP$_{tgt}$ on target text.** Given parallel sequences, we take the ratio between the length in tokens of the target text with our tokenizer and the reference length.

**Rényi efficiency ratio with respect to SP$_{tgt}$.** This is analogous to the length ratio but for the Rényi efficiency, which is an entropy-based measure that quantifies deviation from a uniform distribution. Zouhar et al. (2023) show that this metric correlates well with BLEU scores (Papineni et al., 2002) in machine translation.

**Begin of word.** We count the proportion of tokens in the vocabulary that represent the beginning of a word.

For the alignment task, we compute the additional metrics:

**Eflomal score.** Hämmerl et al. (2025) show that this correlates with cross-lingual transfer. It measures the "maximum unnormalized log-probability of links in the last sampling iteration" (Vázquez et al., 2019).

## D. SacreBleu and COMET

We use the following settings for SacreBleu:
```
BLEU|nrefs:1|case:mixed|eff:no|tok:13a|smooth:exp|version:2.4.2
chrF2++|nrefs:1|case:mixed|eff:yes|nc:6|nw:2|space:no|version:2.4.2
TER|nrefs:1|case:lc|tok:tercom|norm:no|punct:yes|asian:no|version:2.4.2
```

And Unbabel/wmt22-comet-da for computing COMET. Note that this model is not trained for Maltese or Upper Sorbian.

## E. Tokenizer Pseudo-code

Algorithm 1 summarizes the training loop. Many details regarding the settings of the tokenizer (e.g., number of iterations, character coverage, ...) are left out for simplicity.

---

**Algorithm 1** Training algorithm.

---

    **function** TRAIN($src, trg$)
**input**     $src$: list of tokenized source sentences, $trg$: list of target sentences
**output**    $c$: co-occurence table, $\mathcal{V}$: vocabulary
        $c \leftarrow 0$
        $\mathcal{V} \leftarrow \{\}$
        {Initialize the co-occurrence table}
        **for all** $(S, T) \in (src, trg)$ **do**
            **for all** $(t, s) \in \text{SPANS}(T) \times S$ **do**
                $c(t, s) \leftarrow c(t, s) + 1$
                $\mathcal{V} \leftarrow \mathcal{V} \cup \{t\}$
            **end for**
        **end for**
        {Training iterations}
        **for** $i \leftarrow 1$ **to** $n_{iterations}$ **do**
            $c \leftarrow \text{COUNT}(c, \mathcal{V}, src, trg)$
            **if** $i \mod n_{subiterations} == 0$ **then**
                {Remove the target tokens with the lowest $I(t, V_{\text{src}})$}
                $\mathcal{V} \leftarrow \text{PRUNE}(V)$
            **end if**
        **end for**
        {Update the table with the final vocabulary.}
        $c \leftarrow \text{COUNT}(c, \mathcal{V}, src, trg)$
        **return** $c, \mathcal{V}$
    **end function**

    **function** COUNT($c, \mathcal{V}, src, trg$)
        $c_{new} \leftarrow 0$
        **for all** $(T) \in (src, trg)$ **do**
            **for all** $(t, s) \in \text{SPANS}(T) \times T$ **do**
                **if** $t \in \mathcal{V}$ **then**
                    $pref \leftarrow \text{PREFIX}(T, t)$
                    $suff \leftarrow \text{SUFFIX}(T, t)$
                    {SCORE(...) computes the conditional probability of a token given a sentence and the co-occurrence table.}
                    {$\text{SCORE}_{tok}$(...) is similar, marginalize over the possible tokenizations of the prefix or suffix.}
                    $c_{new}(t, s) \leftarrow c_{new}(t, s) + \dfrac{\text{SCORE}(c, t, S)\text{SCORE}_{tok}(c, pref, S)\text{SCORE}_{tok}(c, suff, S)}{\text{length}(src)}$
                **end if**
            **end for**
        **end for**
        **return** $c_{new}$
    **end function**

---

---
**Algorithm 2** Tokenization algorithm, adapted from Unigram.

---

**function** $\textsc{Tokenize}(c, src, trg)$
  $\{\sim$ Forward pass from Unigram$\}$
  **for all** $(S, T) \in (src, trg)$ **do**
    $best \leftarrow [0, -\infty, ...]$
    $sizes \leftarrow [0, 0, ...]$
    $\{$Iterate over the spans starting from $i\}$
    **for** $i \leftarrow 1, \text{length}(T) + 1$ **do**
      **for** $j \leftarrow i - 1, -1$ **do**
        $t \leftarrow T[j : i]$
        $\{t$ is in the vocabulary$\}$
        **if** $c(t, :) \neq 0$ **then**
          $score \leftarrow p(t \mid S)$
          $\{$Store the loss and size of the token$\}$
          **if** $(best[j] + score) > best[i]$ **then**
            $best[i] \leftarrow best[j] + score$
            $sizes[i] \leftarrow i - j$
          **end if**
        **end if**
      **end for**
    **end for**
    $\{\sim$ Backward pass from Unigram$\}$
    $\mathcal{L} \leftarrow best[-1]$
    $i \leftarrow \ell(sizes)$
    $tok \leftarrow []$
    $\{$Add tokens with size from $sizes\}$
    **while** $i > 1$ **do**
      $next \leftarrow i - sizes[i - 1]$
      $\textsc{Append}(tok, T[next - 1 : i - 1])$
      $i \leftarrow next$
    **end while**
    **yield** $\textsc{Reverse}(tok)$
  **end for**
**end function**

---

# F. Machine Translation Hyperparameters

| Model options | | Validation options | |
|---|---|---|---|
| type | transformer | beam-size | 8 |
| dim-emb | 512 | normalize | 0.6 |
| enc-depth | 6 | valid-freq | 100ku |
| dec-depth | 6 | | ce-mean-words |
| tied-embeddings | true | | bleu |
| transformer-heads | 8 | valid-metrics | perplexity |
| transformer-dim-ffn | 2048 | | translation |
| transformer-ffn-activation | relu | | chrf |
| transformer-preprocess | "" | valid-mini-batch | 16 |
| transformer-postprocess | dan | | |
| transformer-dropout | 0.1 | | |

| Training options | | | |
|---|---|---|---|
| cost-type | ce-mean-words | lr-warmup | 16000 |
| max-length | 512 | lr-report | true |
| mini-batch | 1000 | label-smoothing | 0.1 |
| mini-batch-fit | true | clip-norm | 1 |
| maxi-batch | 1000 | exponential-smoothing | 0.0001 |
| optimizer-params | [0.9, 0.98, 1e-09] | disp-freq | 10ku |
| sync-sgd | true | early-stopping | 10 |
| learn-rate | 0.0003 | after | 1Mu |
| lr-decay-inv-sqrt | 16000 | shuffle-in-ram | true |

# G. Language Modelling Hyperparameters

| Model options | | Training options | |
|---|---|---|---|
| activation_function | gelu_new | per_device_train_batch_size | 64 |
| attn_pdrop | 0.1 | per_device_eval_batch_size | 64 |
| embd_pdrop | 0.1 | gradient_accumulation_steps | 8 |
| initializer_range | 0.02 | max_steps | 2_000_000 / 64 |
| layer_norm_epsilon | 1e-05 | weight_decay | 0.1 |
| model_type | gpt2 | warmup_steps | 1_000 |
| n_embd | 768 | lr_scheduler_type | cosine |
| n_head | 12 | learning_rate | 5e-5 |
| n_inner | null | fp16 | True |
| n_layer | 12 | | |
| n_positions | 512 | | |
| reorder_and_upcast_attn | false | | |
| resid_pdrop | 0.1 | | |
| scale_attn_by_inverse_layer_idx | false | | |
| scale_attn_weights | true | | |
| transformers_version | 4.51.2 | | |
| use_cache | true | | |
| vocab_size | 8000/16000/32000 | | |

## H. Intrinsic Evaluation

Table 10: Additional parity scores from the models trained on the smaller training sets.

| | | Parity (↓) | | | | | |
|---|---|---|---|---|---|---|---|
| Size | Model | 8k | 16k | 32k | 8k | 16k | 32k |
| | | Fra → Ita | | | Ita → Fra | | |
| 100k | PairedSP | 1.26 | 1.15 | 1.03 | 1.19 | 1.15 | 1.10 |
| | PairedSP$_M$ | 1.21 | 1.09 | 1.00 | 1.15 | 1.11 | 1.11 |
| | PairedSP$_{EM}$ | 1.04 | 1.02 | 0.98 | 1.10 | 1.08 | 1.06 |
| | SP$_{tgt}$ | 0.99 | 0.98 | 0.98 | 1.01 | 1.02 | 1.02 |
| 500k | PairedSP | 1.24 | 1.11 | 1.05 | 1.21 | 1.15 | 1.12 |
| | PairedSP$_M$ | 1.21 | 1.07 | 0.99 | 1.17 | 1.11 | 1.07 |
| | PairedSP$_{EM}$ | 1.07 | 1.05 | 1.03 | 1.14 | 1.14 | 1.13 |
| | SP$_{tgt}$ | 0.97 | 0.96 | 0.96 | 1.03 | 1.04 | 1.04 |
| | | Ces → Ukr | | | Ukr → Ces | | |
| 100k | PairedSP | 1.57 | 1.48 | 1.37 | 1.32 | 1.24 | 1.13 |
| | PairedSP$_M$ | 1.56 | 1.45 | 1.33 | 1.28 | 1.20 | 1.10 |
| | PairedSP$_{EM}$ | 1.09 | 1.10 | 1.08 | 0.98 | 1.00 | 0.97 |
| | SP$_{tgt}$ | 1.07 | 1.06 | 1.06 | 0.93 | 0.95 | 0.94 |
| 500k | PairedSP | 1.57 | 1.48 | 1.38 | 1.39 | 1.28 | 1.17 |
| | PairedSP$_M$ | 1.56 | 1.46 | 1.34 | 1.37 | 1.24 | 1.11 |
| | PairedSP$_{EM}$ | 1.09 | 1.13 | 1.13 | 1.04 | 1.05 | 1.03 |
| | SP$_{tgt}$ | 1.02 | 1.03 | 1.05 | 0.98 | 0.97 | 0.95 |

Table 11: Additional fertility scores from the models trained on the smaller training sets.

| | | Fertiliy (↓) | | | | | |
|---|---|---|---|---|---|---|---|
| Size | Model | 8k | 16k | 32k | 8k | 16k | 32k |
| | | Fra → Ita | | | Ita → Fra | | |
| 100k | PairedSP | 1.84 | 1.56 | 1.34 | 1.57 | 1.40 | 1.29 |
| | PairedSP$_M$ | 1.78 | 1.48 | 1.31 | 1.52 | 1.36 | 1.30 |
| | PairedSP$_{EM}$ | 1.52 | 1.39 | 1.28 | 1.45 | 1.32 | 1.24 |
| | SP$_{tgt}$ | 1.45 | 1.34 | 1.28 | 1.34 | 1.24 | 1.19 |
| 500k | PairedSP | 1.77 | 1.45 | 1.30 | 1.52 | 1.32 | 1.21 |
| | PairedSP$_M$ | 1.72 | 1.39 | 1.23 | 1.47 | 1.27 | 1.16 |
| | PairedSP$_{EM}$ | 1.52 | 1.37 | 1.27 | 1.44 | 1.30 | 1.22 |
| | SP$_{tgt}$ | 1.38 | 1.25 | 1.19 | 1.30 | 1.19 | 1.13 |
| | | Ces → Ukr | | | Ukr → Ces | | |
| 100k | PairedSP | 2.66 | 2.24 | 1.90 | 2.49 | 2.06 | 1.74 |
| | PairedSP$_M$ | 2.64 | 2.19 | 1.84 | 2.42 | 2.00 | 1.69 |
| | PairedSP$_{EM}$ | 1.84 | 1.66 | 1.49 | 1.85 | 1.67 | 1.49 |
| | SP$_{tgt}$ | 1.81 | 1.60 | 1.47 | 1.76 | 1.58 | 1.45 |
| 500k | PairedSP | 2.60 | 2.14 | 1.78 | 2.45 | 1.99 | 1.65 |
| | PairedSP$_M$ | 2.58 | 2.11 | 1.73 | 2.40 | 1.93 | 1.57 |
| | PairedSP$_{EM}$ | 1.81 | 1.63 | 1.46 | 1.83 | 1.64 | 1.46 |
| | SP$_{tgt}$ | 1.69 | 1.49 | 1.36 | 1.72 | 1.51 | 1.35 |

Table 12: Single Character tokens in the tokenized text.

| | | Single Character (↓) | | | | | |
|---|---|---|---|---|---|---|---|
| Size | Model | 8k | 16k | 32k | 8k | 16k | 32k |
| | | Fra → Ita | | | Ita → Fra | | |
| 100k | PairedSP | 0.34 | 0.23 | 0.13 | 0.23 | 0.17 | 0.13 |
| | PairedSP$_M$ | 0.31 | 0.18 | 0.10 | 0.21 | 0.14 | 0.13 |
| | PairedSP$_{EM}$ | 0.09 | 0.07 | 0.06 | 0.09 | 0.07 | 0.06 |
| | SP$_{tgt}$ | 0.14 | 0.13 | 0.12 | 0.11 | 0.09 | 0.07 |
| 500k | PairedSP | 0.32 | 0.17 | 0.12 | 0.24 | 0.14 | 0.10 |
| | PairedSP$_M$ | 0.30 | 0.14 | 0.07 | 0.21 | 0.11 | 0.06 |
| | PairedSP$_{EM}$ | 0.09 | 0.07 | 0.06 | 0.10 | 0.07 | 0.06 |
| | SP$_{tgt}$ | 0.09 | 0.08 | 0.07 | 0.09 | 0.06 | 0.05 |
| 1M | PairedSP | 0.32 | 0.17 | 0.11 | 0.25 | 0.14 | 0.09 |
| | PairedSP$_M$ | 1.00 | 0.13 | 0.07 | 0.23 | 0.10 | 0.05 |
| | PairedSP$_{EM}$ | 0.09 | 0.07 | 0.06 | 0.10 | 0.08 | 0.06 |
| | SP$_{tgt}$ | 0.07 | 0.06 | 0.05 | 0.09 | 0.06 | 0.04 |
| | | Ces → Ukr | | | Ukr → Ces | | |
| 100k | PairedSP | 0.59 | 0.49 | 0.39 | 0.52 | 0.39 | 0.30 |
| | PairedSP$_M$ | 0.59 | 0.48 | 0.37 | 0.50 | 0.36 | 0.28 |
| | PairedSP$_{EM}$ | 0.12 | 0.10 | 0.09 | 0.13 | 0.11 | 0.09 |
| | SP$_{tgt}$ | 0.24 | 0.19 | 0.16 | 0.22 | 0.19 | 0.16 |
| 500k | PairedSP | 0.59 | 0.48 | 0.36 | 0.53 | 0.39 | 0.28 |
| | PairedSP$_M$ | 0.58 | 0.47 | 0.34 | 0.51 | 0.36 | 0.23 |
| | PairedSP$_{EM}$ | 0.11 | 0.10 | 0.08 | 0.12 | 0.10 | 0.08 |
| | SP$_{tgt}$ | 0.16 | 0.12 | 0.10 | 0.19 | 0.15 | 0.12 |
| 1M | PairedSP | 0.59 | 0.48 | 0.36 | 0.53 | 0.40 | 0.27 |
| | PairedSP$_M$ | 0.59 | 0.47 | 0.34 | 0.52 | 0.37 | 0.23 |
| | PairedSP$_{EM}$ | 0.11 | 0.10 | 0.08 | 0.13 | 0.10 | 0.08 |
| | SP$_{tgt}$ | 0.15 | 0.11 | 0.09 | 0.18 | 0.14 | 0.11 |
| | | Ita → Mlt | | | Mlt → Ita | | |
| 100k | PairedSP | 0.41 | 0.24 | 0.16 | 0.38 | 0.24 | 0.16 |
| | PairedSP$_M$ | 0.41 | 0.22 | 0.13 | 0.36 | 0.21 | 0.12 |
| | PairedSP$_{EM}$ | 0.09 | 0.08 | 0.06 | 0.09 | 0.08 | 0.06 |
| | SP$_{tgt}$ | 0.10 | 0.08 | 0.07 | 0.12 | 0.12 | 0.11 |
| | | Deu → Hsb | | | Hsb → Deu | | |
| 60k | PairedSP | 0.52 | 0.37 | 0.27 | 0.42 | 0.30 | 0.14 |
| | PairedSP$_M$ | 0.51 | 0.35 | 0.25 | 0.42 | 0.30 | 0.19 |
| | PairedSP$_{EM}$ | 0.10 | 0.08 | 0.07 | 0.09 | 0.08 | 0.05 |
| | SP$_{tgt}$ | 0.20 | 0.18 | 0.15 | 0.19 | 0.16 | 0.12 |

Table 13: Vocabulary usage of the different tokenizers.

| | | Vocabulary Usage (↑) | | | | | |
|---|---|---|---|---|---|---|---|
| Size | Model | 8k | 16k | 32k | 8k | 16k | 32k |
| | | **Fra → Ita** | | | **Ita → Fra** | | |
| 100k | PairedSP | 0.43 | 0.28 | 0.18 | 0.43 | 0.28 | 0.17 |
| | PairedSP$_M$ | 0.43 | 0.29 | 0.17 | 0.43 | 0.28 | 0.16 |
| | PairedSP$_{EM}$ | 0.52 | 0.35 | 0.22 | 0.48 | 0.33 | 0.20 |
| | SP$_{tgt}$ | 0.57 | 0.36 | 0.21 | 0.55 | 0.34 | 0.19 |
| 500k | PairedSP | 0.44 | 0.30 | 0.18 | 0.46 | 0.30 | 0.18 |
| | PairedSP$_M$ | 0.45 | 0.31 | 0.19 | 0.47 | 0.31 | 0.19 |
| | PairedSP$_{EM}$ | 0.52 | 0.36 | 0.21 | 0.50 | 0.34 | 0.20 |
| | SP$_{tgt}$ | 0.57 | 0.38 | 0.21 | 0.56 | 0.36 | 0.20 |
| 1M | PairedSP | 0.45 | 0.31 | 0.19 | 0.46 | 0.31 | 0.19 |
| | PairedSP$_M$ | 0.01 | 0.31 | 0.19 | 0.47 | 0.32 | 0.19 |
| | PairedSP$_{EM}$ | 0.52 | 0.36 | 0.21 | 0.49 | 0.34 | 0.20 |
| | SP$_{tgt}$ | 0.56 | 0.38 | 0.22 | 0.56 | 0.37 | 0.20 |
| | | **Ces → Ukr** | | | **Ukr → Ces** | | |
| 100k | PairedSP | 0.40 | 0.28 | 0.18 | 0.41 | 0.28 | 0.18 |
| | PairedSP$_M$ | 0.41 | 0.28 | 0.18 | 0.42 | 0.29 | 0.19 |
| | PairedSP$_{EM}$ | 0.54 | 0.38 | 0.24 | 0.54 | 0.38 | 0.24 |
| | SP$_{tgt}$ | 0.56 | 0.38 | 0.23 | 0.58 | 0.39 | 0.23 |
| 500k | PairedSP | 0.42 | 0.30 | 0.19 | 0.42 | 0.30 | 0.20 |
| | PairedSP$_M$ | 0.43 | 0.30 | 0.19 | 0.43 | 0.31 | 0.20 |
| | PairedSP$_{EM}$ | 0.55 | 0.39 | 0.25 | 0.55 | 0.39 | 0.25 |
| | SP$_{tgt}$ | 0.58 | 0.41 | 0.24 | 0.58 | 0.41 | 0.25 |
| 1M | PairedSP | 0.42 | 0.30 | 0.19 | 0.42 | 0.30 | 0.20 |
| | PairedSP$_M$ | 0.43 | 0.30 | 0.20 | 0.42 | 0.31 | 0.20 |
| | PairedSP$_{EM}$ | 0.53 | 0.39 | 0.25 | 0.54 | 0.39 | 0.25 |
| | SP$_{tgt}$ | 0.58 | 0.42 | 0.25 | 0.58 | 0.41 | 0.25 |
| | | **Ita → Mlt** | | | **Mlt → Ita** | | |
| 100k | PairedSP | 0.39 | 0.27 | 0.16 | 0.42 | 0.28 | 0.18 |
| | PairedSP$_M$ | 0.39 | 0.27 | 0.17 | 0.42 | 0.29 | 0.18 |
| | PairedSP$_{EM}$ | 0.50 | 0.34 | 0.21 | 0.51 | 0.34 | 0.21 |
| | SP$_{tgt}$ | 0.54 | 0.35 | 0.20 | 0.55 | 0.37 | 0.21 |
| | | **Deu → Hsb** | | | **Hsb → Deu** | | |
| 60k | PairedSP | 0.47 | 0.32 | 0.20 | 0.42 | 0.29 | 0.18 |
| | PairedSP$_M$ | 0.47 | 0.32 | 0.20 | 0.42 | 0.29 | 0.17 |
| | PairedSP$_{EM}$ | 0.55 | 0.38 | 0.24 | 0.50 | 0.33 | 0.21 |
| | SP$_{tgt}$ | 0.60 | 0.39 | 0.23 | 0.58 | 0.37 | 0.21 |

Table 14: Vocabulary overlap with SP$_{tgt}$.

| | | Vocabulary Overlap | | | | | |
|---|---|---|---|---|---|---|---|
| Size | Model | 8k | 16k | 32k | 8k | 16k | 32k |
| | | **Fra → Ita** | | | **Ita → Fra** | | |
| 100k | PairedSP | 0.42 | 0.39 | 0.40 | 0.52 | 0.49 | 0.47 |
| | PairedSP$_M$ | 0.42 | 0.39 | 0.40 | 0.52 | 0.49 | 0.47 |
| | PairedSP$_{EM}$ | 0.34 | 0.29 | 0.31 | 0.34 | 0.30 | 0.32 |
| 500k | PairedSP | 0.58 | 0.56 | 0.50 | 0.64 | 0.62 | 0.57 |
| | PairedSP$_M$ | 0.58 | 0.56 | 0.50 | 0.64 | 0.62 | 0.57 |
| | PairedSP$_{EM}$ | 0.39 | 0.34 | 0.29 | 0.40 | 0.35 | 0.29 |
| 1M | PairedSP | 0.63 | 0.62 | 0.57 | 0.67 | 0.67 | 0.63 |
| | PairedSP$_M$ | 0.63 | 0.62 | 0.57 | 0.67 | 0.67 | 0.63 |
| | PairedSP$_{EM}$ | 0.42 | 0.35 | 0.30 | 0.42 | 0.36 | 0.30 |
| | | **Ces → Ukr** | | | **Ukr → Ces** | | |
| 100k | PairedSP | 0.39 | 0.38 | 0.34 | 0.46 | 0.43 | 0.39 |
| | PairedSP$_M$ | 0.39 | 0.38 | 0.34 | 0.46 | 0.43 | 0.39 |
| | PairedSP$_{EM}$ | 0.32 | 0.26 | 0.25 | 0.34 | 0.27 | 0.26 |
| 500k | PairedSP | 0.47 | 0.47 | 0.46 | 0.54 | 0.54 | 0.52 |
| | PairedSP$_M$ | 0.47 | 0.47 | 0.46 | 0.54 | 0.54 | 0.52 |
| | PairedSP$_{EM}$ | 0.41 | 0.33 | 0.29 | 0.43 | 0.34 | 0.31 |
| 1M | PairedSP | 0.51 | 0.51 | 0.51 | 0.56 | 0.57 | 0.57 |
| | PairedSP$_M$ | 0.51 | 0.51 | 0.51 | 0.56 | 0.57 | 0.57 |
| | PairedSP$_{EM}$ | 0.43 | 0.34 | 0.31 | 0.46 | 0.36 | 0.32 |
| | | **Ita → Mlt** | | | **Mlt → Ita** | | |
| 100k | PairedSP | 0.50 | 0.47 | 0.40 | 0.47 | 0.43 | 0.37 |
| | PairedSP$_M$ | 0.50 | 0.47 | 0.40 | 0.47 | 0.43 | 0.37 |
| | PairedSP$_{EM}$ | 0.38 | 0.30 | 0.29 | 0.35 | 0.28 | 0.29 |
| | | **Deu → Hsb** | | | **Hsb → Deu** | | |
| 60k | PairedSP | 0.33 | 0.33 | 0.31 | 0.42 | 0.39 | 0.35 |
| | PairedSP$_M$ | 0.33 | 0.33 | 0.31 | 0.42 | 0.39 | 0.35 |
| | PairedSP$_{EM}$ | 0.26 | 0.21 | 0.24 | 0.28 | 0.23 | 0.30 |

Table 15: Lnegth ratio between PairedSP (and derived models) and SP$_{tgt}$.

| | | Length Ratio ($\downarrow$) | | | | | |
|---|---|---|---|---|---|---|---|
| Size | Model | 8k | 16k | 32k | 8k | 16k | 32k |
| | | Fra $\rightarrow$ Ita | | | Ita $\rightarrow$ Fra | | |
| 100k | PairedSP | 1.27 | 1.17 | 1.05 | 1.17 | 1.13 | 1.08 |
| | PairedSP$_M$ | 1.23 | 1.11 | 1.02 | 1.14 | 1.09 | 1.09 |
| | PairedSP$_{EM}$ | 1.05 | 1.04 | 1.00 | 1.08 | 1.06 | 1.04 |
| 500k | PairedSP | 1.28 | 1.15 | 1.09 | 1.17 | 1.11 | 1.07 |
| | PairedSP$_M$ | 1.25 | 1.11 | 1.04 | 1.13 | 1.07 | 1.03 |
| | PairedSP$_{EM}$ | 1.10 | 1.09 | 1.07 | 1.11 | 1.09 | 1.08 |
| 1M | PairedSP | 1.29 | 1.16 | 1.09 | 1.17 | 1.10 | 1.07 |
| | PairedSP$_M$ | 4.11 | 1.12 | 1.04 | 1.14 | 1.06 | 1.03 |
| | PairedSP$_{EM}$ | 1.10 | 1.12 | 1.10 | 1.11 | 1.11 | 1.10 |
| | | Ces $\rightarrow$ Ukr | | | Ukr $\rightarrow$ Ces | | |
| 100k | PairedSP | 1.47 | 1.40 | 1.29 | 1.41 | 1.31 | 1.20 |
| | PairedSP$_M$ | 1.46 | 1.37 | 1.25 | 1.37 | 1.27 | 1.16 |
| | PairedSP$_{EM}$ | 1.02 | 1.04 | 1.01 | 1.05 | 1.06 | 1.03 |
| 500k | PairedSP | 1.54 | 1.44 | 1.31 | 1.42 | 1.32 | 1.22 |
| | PairedSP$_M$ | 1.53 | 1.42 | 1.28 | 1.39 | 1.28 | 1.16 |
| | PairedSP$_{EM}$ | 1.07 | 1.10 | 1.08 | 1.06 | 1.09 | 1.08 |
| 1M | PairedSP | 1.56 | 1.46 | 1.33 | 1.44 | 1.34 | 1.24 |
| | PairedSP$_M$ | 1.55 | 1.44 | 1.30 | 1.42 | 1.30 | 1.18 |
| | PairedSP$_{EM}$ | 1.09 | 1.11 | 1.10 | 1.07 | 1.10 | 1.10 |
| | | Ita $\rightarrow$ Mlt | | | Mlt $\rightarrow$ Ita | | |
| 100k | PairedSP | 1.33 | 1.18 | 1.10 | 1.31 | 1.18 | 1.09 |
| | PairedSP$_M$ | 1.31 | 1.15 | 1.06 | 1.28 | 1.13 | 1.03 |
| | PairedSP$_{EM}$ | 1.08 | 1.08 | 1.04 | 1.07 | 1.07 | 1.03 |
| | | Deu $\rightarrow$ Hsb | | | Hsb $\rightarrow$ Deu | | |
| 60k | PairedSP | 1.37 | 1.21 | 1.10 | 1.32 | 1.19 | 1.02 |
| | PairedSP$_M$ | 1.35 | 1.19 | 1.07 | 1.32 | 1.17 | 1.06 |
| | PairedSP$_{EM}$ | 1.03 | 1.02 | 0.97 | 1.05 | 1.04 | 0.99 |

Table 16: Ratio between the Rényi efficiency of PairedSP (and derived models) and SP$_{tgt}$

| | | Rényi Ratio ($\uparrow$) | | | | | |
|---|---|---|---|---|---|---|---|
| Size | Model | 8k | 16k | 32k | 8k | 16k | 32k |
| | | Fra $\rightarrow$ Ita | | | Ita $\rightarrow$ Fra | | |
| 100k | PairedSP | 0.92 | 0.95 | 0.99 | 0.95 | 0.97 | 0.98 |
| | PairedSP$_M$ | 0.93 | 0.97 | 1.00 | 0.96 | 0.98 | 0.98 |
| | PairedSP$_{EM}$ | 0.98 | 0.99 | 1.00 | 0.98 | 0.98 | 0.99 |
| 500k | PairedSP | 0.92 | 0.96 | 0.98 | 0.95 | 0.97 | 0.98 |
| | PairedSP$_M$ | 0.93 | 0.97 | 0.99 | 0.96 | 0.98 | 0.99 |
| | PairedSP$_{EM}$ | 0.97 | 0.98 | 0.98 | 0.97 | 0.98 | 0.98 |
| 1M | PairedSP | 0.92 | 0.96 | 0.98 | 0.95 | 0.98 | 0.99 |
| | PairedSP$_M$ | 0.47 | 0.97 | 0.99 | 0.96 | 0.99 | 0.99 |
| | PairedSP$_{EM}$ | 0.97 | 0.97 | 0.98 | 0.97 | 0.97 | 0.98 |
| | | Ces $\rightarrow$ Ukr | | | Ukr $\rightarrow$ Ces | | |
| 100k | PairedSP | 0.83 | 0.86 | 0.91 | 0.85 | 0.89 | 0.93 |
| | PairedSP$_M$ | 0.83 | 0.87 | 0.92 | 0.86 | 0.91 | 0.95 |
| | PairedSP$_{EM}$ | 0.99 | 0.99 | 1.00 | 0.98 | 0.98 | 0.99 |
| 500k | PairedSP | 0.82 | 0.86 | 0.91 | 0.85 | 0.89 | 0.93 |
| | PairedSP$_M$ | 0.82 | 0.87 | 0.92 | 0.86 | 0.90 | 0.95 |
| | PairedSP$_{EM}$ | 0.97 | 0.97 | 0.98 | 0.98 | 0.97 | 0.98 |
| 1M | PairedSP | 0.81 | 0.86 | 0.91 | 0.84 | 0.89 | 0.93 |
| | PairedSP$_M$ | 0.81 | 0.86 | 0.91 | 0.85 | 0.90 | 0.95 |
| | PairedSP$_{EM}$ | 0.97 | 0.96 | 0.97 | 0.97 | 0.96 | 0.97 |
| | | Ita $\rightarrow$ Mlt | | | Mlt $\rightarrow$ Ita | | |
| 100k | PairedSP | 0.90 | 0.95 | 0.97 | 0.91 | 0.95 | 0.98 |
| | PairedSP$_M$ | 0.90 | 0.96 | 0.98 | 0.92 | 0.97 | 0.99 |
| | PairedSP$_{EM}$ | 0.98 | 0.98 | 0.99 | 0.98 | 0.98 | 0.99 |
| | | Deu $\rightarrow$ Hsb | | | Hsb $\rightarrow$ Deu | | |
| 60k | PairedSP | 0.87 | 0.93 | 0.97 | 0.91 | 0.95 | 1.00 |
| | PairedSP$_M$ | 0.88 | 0.94 | 0.98 | 0.91 | 0.96 | 0.99 |
| | PairedSP$_{EM}$ | 0.99 | 0.99 | 1.01 | 0.99 | 0.99 | 1.00 |

Table 17: Begin-of-word tokens in the vocabulary.

| Size | Model | 8k | 16k | 32k | 8k | 16k | 32k |
|---|---|---|---|---|---|---|---|
| | | **Start Word** | | | | | |
| | | **Fra → Ita** | | | **Ita → Fra** | | |
| 100k | PairedSP | 0.95 | 0.97 | 0.90 | 0.91 | 0.90 | 0.80 |
| | PairedSP$_M$ | 0.95 | 0.97 | 0.90 | 0.91 | 0.90 | 0.80 |
| | PairedSP$_{EM}$ | 0.40 | 0.37 | 0.35 | 0.38 | 0.36 | 0.33 |
| | SP$_{tgt}$ | 0.80 | 0.81 | 0.75 | 0.79 | 0.80 | 0.72 |
| 500k | PairedSP | 0.93 | 0.96 | 0.96 | 0.92 | 0.94 | 0.92 |
| | PairedSP$_M$ | 0.93 | 0.96 | 0.96 | 0.92 | 0.94 | 0.92 |
| | PairedSP$_{EM}$ | 0.39 | 0.38 | 0.35 | 0.39 | 0.38 | 0.35 |
| | SP$_{tgt}$ | 0.79 | 0.84 | 0.84 | 0.79 | 0.83 | 0.82 |
| 1M | PairedSP | 0.91 | 0.95 | 0.96 | 0.90 | 0.93 | 0.92 |
| | PairedSP$_M$ | 0.91 | 0.95 | 0.96 | 0.90 | 0.93 | 0.92 |
| | PairedSP$_{EM}$ | 0.39 | 0.37 | 0.36 | 0.38 | 0.37 | 0.34 |
| | SP$_{tgt}$ | 0.78 | 0.84 | 0.85 | 0.76 | 0.82 | 0.84 |
| | | **Ces → Ukr** | | | **Ukr → Ces** | | |
| 100k | PairedSP | 0.95 | 0.97 | 0.98 | 0.94 | 0.94 | 0.95 |
| | PairedSP$_M$ | 0.95 | 0.97 | 0.98 | 0.94 | 0.94 | 0.95 |
| | PairedSP$_{EM}$ | 0.39 | 0.35 | 0.34 | 0.41 | 0.37 | 0.36 |
| | SP$_{tgt}$ | 0.76 | 0.81 | 0.80 | 0.79 | 0.84 | 0.82 |
| 500k | PairedSP | 0.93 | 0.96 | 0.98 | 0.93 | 0.94 | 0.96 |
| | PairedSP$_M$ | 0.93 | 0.96 | 0.98 | 0.93 | 0.94 | 0.96 |
| | PairedSP$_{EM}$ | 0.40 | 0.37 | 0.36 | 0.42 | 0.38 | 0.38 |
| | SP$_{tgt}$ | 0.73 | 0.80 | 0.84 | 0.76 | 0.83 | 0.87 |
| 1M | PairedSP | 0.91 | 0.95 | 0.97 | 0.91 | 0.94 | 0.95 |
| | PairedSP$_M$ | 0.91 | 0.95 | 0.97 | 0.91 | 0.94 | 0.95 |
| | PairedSP$_{EM}$ | 0.39 | 0.36 | 0.36 | 0.42 | 0.38 | 0.38 |
| | SP$_{tgt}$ | 0.71 | 0.79 | 0.84 | 0.73 | 0.81 | 0.86 |
| | | **Ita → Mlt** | | | **Mlt → Ita** | | |
| 100k | PairedSP | 0.95 | 0.94 | 0.92 | 0.95 | 0.96 | 0.96 |
| | PairedSP$_M$ | 0.95 | 0.94 | 0.92 | 0.95 | 0.96 | 0.96 |
| | PairedSP$_{EM}$ | 0.39 | 0.35 | 0.35 | 0.40 | 0.36 | 0.36 |
| | SP$_{tgt}$ | 0.73 | 0.75 | 0.73 | 0.81 | 0.83 | 0.80 |
| | | **Deu → Hsb** | | | **Hsb → Deu** | | |
| 60k | PairedSP | 0.98 | 0.98 | 0.99 | 0.96 | 0.95 | 0.84 |
| | PairedSP$_M$ | 0.98 | 0.98 | 0.99 | 0.96 | 0.95 | 0.84 |
| | PairedSP$_{EM}$ | 0.38 | 0.34 | 0.34 | 0.37 | 0.31 | 0.31 |
| | SP$_{tgt}$ | 0.85 | 0.88 | 0.84 | 0.71 | 0.73 | 0.66 |

Table 18: Eflomal scores of the aligned text.

| Size | Model | 8k | 16k | 32k | 8k | 16k | 32k |
|---|---|---|---|---|---|---|---|
| | | **Eflomal scores ($\downarrow$)** | | | | | |
| | | **Fra → Ita** | | | **Ita → Fra** | | |
| 1M | PairedSP | 5.61 | 5.54 | 5.36 | 5.36 | 5.25 | 5.08 |
| | SP$_{tgt}$ | 5.07 | 4.99 | 4.89 | 4.93 | 4.82 | 4.75 |
| | | **Ces → Ukr** | | | **Ukr → Ces** | | |
| 1M | PairedSP | 6.54 | 6.96 | 6.98 | 6.43 | 6.70 | 6.64 |
| | SP$_{tgt}$ | 6.02 | 6.08 | 6.03 | 5.90 | 5.98 | 5.89 |
| | | **Ita → Mlt** | | | **Mlt → Ita** | | |
| 100k | PairedSP | 6.10 | 5.97 | 5.89 | 6.11 | 6.31 | 6.12 |
| | SP$_{tgt}$ | 5.46 | 5.39 | 5.36 | 5.51 | 5.55 | 5.43 |
| | | **Deu → Hsb** | | | **Hsb → Deu** | | |
| 60k | PairedSP | 4.61 | 4.52 | 4.14 | 4.70 | 4.29 | 3.84 |
| | SP$_{tgt}$ | 3.47 | 3.34 | 3.16 | 3.53 | 3.58 | 3.35 |

# I. Machine Translation Evaluation

Table 19: Average BLEU scores on the different language pairs and vocabulary sizes.

| BLEU (↑) | | | | | | |
|---|---|---|---|---|---|---|
| Model | 8k | 16k | 32k | 8k | 16k | 32k |
| | **Fra → Ita** | | | **Ita → Fra** | | |
| SP$_{src}$ + PairedSP | 24.5 | 24.2 | 22.9 | 25.1 | 24.7 | 23.8 |
| SP$_{src}$ + SP$_{tgt}$ | 25.1 | 24.8 | 23.1 | 25.8 | 25.3 | 23.5 |
| | **Ces → Ukr** | | | **Ukr → Ces** | | |
| SP$_{src}$ + PairedSP | 12.5 | 12.4 | 10.6 | 18.9 | 19.1 | 19.2 |
| SP$_{src}$ + SP$_{tgt}$ | 20.0 | 19.8 | 18.8 | 21.6 | 21.2 | 19.6 |
| | **Ita → Mlt** | | | **Mlt → Ita** | | |
| SP$_{src}$ + PairedSP | 0.3 | 5.9 | 5.6 | 12.5 | 18.0 | 17.3 |
| SP$_{src}$ + SP$_{tgt}$ | 6.0 | 5.9 | 5.4 | 18.7 | 18.3 | 17.0 |
| | **Deu → Hsb** | | | **Hsb → Deu** | | |
| SP$_{src}$ + PairedSP | 13.8 | 31.0 | 33.4 | 12.1 | 37.2 | 37.4 |
| SP$_{src}$ + SP$_{tgt}$ | 43.3 | 43.5 | 35.2 | 41.4 | 41.9 | 29.4 |

Table 21: Average Comet scores on the different language pairs and vocabulary sizes. *: Maltese and Upper Sorbian are not included in the Comet training.

| COMET (↑) | | | | | | |
|---|---|---|---|---|---|---|
| Model | 8k | 16k | 32k | 8k | 16k | 32k |
| | **Fra → Ita** | | | **Ita → Fra** | | |
| SP$_{src}$ + PairedSP | 0.797 | 0.799 | 0.786 | 0.750 | 0.753 | 0.741 |
| SP$_{src}$ + SP$_{tgt}$ | 0.801 | 0.805 | 0.780 | 0.755 | 0.760 | 0.737 |
| | **Ces → Ukr** | | | **Ukr → Ces** | | |
| SP$_{src}$ + PairedSP | 0.644 | 0.645 | 0.611 | 0.711 | 0.714 | 0.726 |
| SP$_{src}$ + SP$_{tgt}$ | 0.799 | 0.802 | 0.788 | 0.756 | 0.757 | 0.734 |
| | **Ita → Mlt*** | | | **Mlt* → Ita** | | |
| SP$_{src}$ + PairedSP | 0.436 | 0.590 | 0.591 | 0.521 | 0.624 | 0.609 |
| SP$_{src}$ + SP$_{tgt}$ | 0.592 | 0.592 | 0.592 | 0.634 | 0.634 | 0.607 |
| | **Deu → Hsb*** | | | **Hsb* → Deu** | | |
| SP$_{src}$ + PairedSP | 0.516 | 0.604 | 0.619 | 0.409 | 0.606 | 0.602 |
| SP$_{src}$ + SP$_{tgt}$ | 0.667 | 0.670 | 0.637 | 0.641 | 0.651 | 0.546 |

Table 20: Average TER scores on the different language pairs and vocabulary sizes.

| TER (↓) | | | | | | |
|---|---|---|---|---|---|---|
| Model | 8k | 16k | 32k | 8k | 16k | 32k |
| | **Fra → Ita** | | | **Ita → Fra** | | |
| SP$_{src}$ + PairedSP | 78.9 | 79.4 | 80.3 | 82.5 | 83.2 | 84.4 |
| SP$_{src}$ + SP$_{tgt}$ | 78.6 | 78.9 | 81.0 | 82.2 | 82.6 | 85.4 |
| | **Ces → Ukr** | | | **Ukr → Ces** | | |
| SP$_{src}$ + PairedSP | 105.4 | 115.6 | 119.6 | 86.5 | 86.7 | 86.2 |
| SP$_{src}$ + SP$_{tgt}$ | 89.7 | 90.2 | 92.4 | 84.4 | 84.8 | 87.0 |
| | **Ita → Mlt** | | | **Mlt → Ita** | | |
| SP$_{src}$ + PairedSP | 317.3 | 138.7 | 140.9 | 137.8 | 84.9 | 86.1 |
| SP$_{src}$ + SP$_{tgt}$ | 138.6 | 139.6 | 143.1 | 84.4 | 84.9 | 86.9 |
| | **Deu → Hsb** | | | **Hsb → Deu** | | |
| SP$_{src}$ + PairedSP | 85.0 | 71.6 | 71.8 | 94.8 | 65.3 | 67.6 |
| SP$_{src}$ + SP$_{tgt}$ | 59.7 | 60.0 | 71.6 | 61.7 | 61.6 | 78.9 |

