# OpenReview forum: "Conditional Unigram Tokenization with Parallel Data"
_ICML.cc/2025/Workshop/TokShop — TokShop_

### Official Review · Reviewer_czdC · 2025-06-04

**Rating:** 7
**Confidence:** 3

**Review:**

The paper proposes to learn a tokenizer for a target language conditioned on the tokenizer of a source language, maximizing cross-lingual alignment. They evaluate the proposed tokenizer using 1) intrinsic evaluation, 2) machine translation and 3) language modelling on 4 language pairs. The evaluations are negative in all but the 3d type of evaluation.

The idea is a nice one and the negative results may be good to know for the community. Most of the experiments make sense to me, the selection of language pairs is decently well motivated. The main weakness is the language modelling comparison. The paper compares the perplexities of language models using different tokenizers. This comparison is invalid, see [Mielke (2019)](https://sjmielke.com/comparing-perplexities.htm). I believe this part of the paper should be revisited or thrown out.

A few more minor improvement suggestions:
* Can you think of alternative names for your models? PairsedSP vs PairedSPm vs PairsedSPem is really not great to visually see the difference.
* l076: establishes -> establish
* l 110-117 - I'm not sure I understand the motivation for using this approach given the mentioned drawbacks - maybe this can be motivated better
* l 152-164 - Alternatively (...) Alternatively ... -> this is a bit hard to parse
* section 4 beginning: I would not call language modelling a "downstream task"

---

### Official Review · Reviewer_b8AS · 2025-06-05
**Novel tokenization method with mixed but important research results**

**Rating:** 7
**Confidence:** 3

**Review:**

The authors propose conditional unigram tokenization, extending SentencePiece-style unigram tokenizers by conditioning each target-language token’s probability on the source-language tokens in a parallel corpus. Instead of using unconditional probabilities p(t), the method uses conditional probabilities p(t|S) where S represents source language tokens. The goal is to improve cross-lingual semantic alignment, particularly benefiting low-resource languages that suffer from overtokenization.

The method was tested across four language pairs (French-Italian, Czech-Ukrainian, Italian-Maltese, German-Upper Sorbian) using both machine translation (MT) and language modeling (LM) tasks. The method shows no improvement in machine translation and mostly underperforms on intrinsic metrics but showed consistent perplexity reduction, suggesting better tokenization for monolingual tasks. The authors hypothesize that the method’s effectiveness is bottlenecked by the quadratic scaling of conditional probability estimation with vocabulary size, making it less data-efficient.

Strengths:
 - Conditioning target token probabilities on source language token for improving tokenization of low resource languages seems novel and proposed for the first time
- The authors perform comprehensive evaluation with the use of multiple language pairs and vocabulary sizes.
 - The authors transparently report negative and mixed results and identify limitations and failure modes.
  - The approach shows consistent improvement in perplexity for language model tasks indicating cross-lingual token counts can help monolingual LMs
- Authors plan to open source code which will help support reproducibility

Weaknesses
- No consistent or significant gains in machine translation tasks which the authors clearly describe
- Quadratic scaling of the conditional probability model hampers performance, especially in low-resource settings
- More complex than standard unigram models in both memory and computational requirements.
- Mostly European languages, could be extended to more diverse pairs
- Practical deployment might be challenging for resource-constrained environments.
- Baseline outperformed the proposed method on standard tokenization metrics like fertility and parity in many cases.

The paper is a solid contribution which improves understanding of cross-lingual tokenization. It provides valuable insights into the challenges and learnings like quadratic scaling bottleneck can help guide future research. It makes a meaningful contribution to the field by clearly showing what doesn't work and why, crucial for further progress.

---

### Decision · Program_Chairs · 2025-06-10

Accept